# Association between the Severity of Nonalcoholic Fatty Liver Disease and the Risk of Coronary Artery Calcification

**DOI:** 10.3390/medicina57080807

**Published:** 2021-08-06

**Authors:** Chien-Chih Chen, Wei-Chien Hsu, Han-Ming Wu, Jiun-Yi Wang, Pei-Yu Yang, I-Ching Lin

**Affiliations:** 1Department of Family Medicine, Asia University Hospital, No. 222, FuSin Rd., Wufeng, Taichung 41354, Taiwan; D51120@auh.org.tw (C.-C.C.); D51121@auh.org.tw (W.-C.H.); D52175@auh.org.tw (H.-M.W.); 2Department of Healthcare Administration, Asia University, No. 500, Lioufeng Rd., Wufeng, Taichung 41354, Taiwan; jjwang@asia.edu.tw; 3Department of Laboratory, Show-Chwan Memorial Hospital, No. 542, Sec1, Chung-Shan Rd., Changhua 50008, Taiwan; a2900@show.org.tw; 4Department of Kinesiology, Health and Leisure, Chienkuo Technology University, No. 1, Chiehshou North Road, Changhua 50000, Taiwan

**Keywords:** nonalcoholic fatty liver disease, coronary artery calcification, subclinical atherosclerosis, cardiac multidetector computed tomography

## Abstract

*Background and Objectives*: There are limited data on the association between severity of non-alcoholic fatty liver disease (NAFLD) and coronary artery calcification. This study investigated sonographic diagnosed NAFLD and coronary artery calcium score (CAC) as detected by cardiac multidetector computed tomography in general populations. *Materials and Methods*: A total of 545 patients were enrolled in this study. NAFLD was diagnosed by ultrasonography examination and CAC score were evaluated by cardiac multidetector computed tomography. The association between NAFLD and artery calcium score stage was determined by logistic regression analysis and Spearman correlation coefficient analysis. *Results*: Of all the participants, 437 (80.2%) had ultrasonography-diagnosed NAFLD and 242 (44%) had coronary artery calcification (CAC > 0). After adjustment for cardiovascular risk factors, the risk of developing coronary artery calcification was 1.36-fold greater in the patients with different severity of NAFLD compared to those without NAFLD (OR = 1.36, 95% CI = 1.07–1.77, *p* = 0.016). The highest OR for separate coronary artery calcification was 1.98 (OR = 1.98, 95% CI = 1.37–2.87, *p* < 0.001) in the left main artery, and the risk was still 1.71-fold greater after adjustments (OR = 1.71, 95% CI = 1.16–2.54, *p* = 0.007). *Conclusions*: This cross-sectional study demonstrated that the severity of NAFLD was associated with the presence of significant coronary artery calcification, especially in the left main coronary artery, suggesting increasing the cardiovascular risk.

## 1. Introduction

Nonalcoholic fatty liver disease (NAFLD) is the most common hepatic disease in developed countries. The estimated global prevalence of NAFLD is 25%, but varies according to the country and region. For example, the prevalence is higher in the Middle East and lower in South Africa, and may be affected by heterogeneous population groups, diet, culture, and different lifestyles [1]. The overall reported prevalence of NAFLD in Asia is 29%, and the incidence has been increasing in the past 20 years in parallel with obesity [2]. NAFLD is expected to become the most important chronic liver disease.

The term “nonalcoholic fatty liver disease” is a pathological diagnosis which includes the histological characteristics of fat accumulation in hepatocytes, hepatocellular necrosis, liver inflammation and fibrosis. A diagnosis of NAFLD must exclude any significant alcohol consumption, autoimmune hepatitis, viral hepatitis and drug-related steatosis [3]. Transabdominal ultrasonography is preferred as the first-line evaluation tool for the diagnosis of NAFLD due to its accessibility, non-invasive procedure and low medical cost [4,5].

Nonalcoholic fatty liver disease is a progressive liver disease associated with obesity, insulin resistance, metabolic syndrome, hypertension and diabetes [1]. Recent studies have reported a strong correlation with cardiovascular events and cardiac mortality [6]. Studies have highlighted the association between NAFLD and coronary artery disease, and there were multiple underlying mechanisms overlapping between NAFLD and arterial calcification [7,8,9]. Stahl et al.’s comprehensive review summarized the pathophysiological mechanisms of NAFLD increasing the risk of cardiovascular disease. It demonstrated six potential pathways, such as systemic inflammation, endothelial dysfunction, hepatic insulin resistance, plaque formation, oxidative stress, and altered lipid metabolism, wound impact the risk of atherosclerosis, cardiovascular mortality in NAFLD populations [10]. The early detection of coronary artery calcification seems to be useful in estimating the risk of cardiovascular diseases. Nevertheless, even meta-analysis revealed “significant NAFLD” with higher risk of incident of fatal and non-fatal CVD [11], but fewer study performed in the relationship between the severity of NAFLD and specific coronary artery calcification. Therefore, the aim of this study was to perform a cross-section analysis of the severity of NAFLD and its relationship with CAC score staging and separated calcified coronary artery involvement in an Asian population.

## 2. Materials and Methods

### 2.1. Study Population

This cross-sectional study initially recruited 938 patients who received 128-slice cardiac multidetector computed tomography (MDCT) at Asia University Hospital from August 2016 to February 2020. There were 10 patients with positive finding for hepatitis B virus, two patients with a positive finding for hepatitis C virus and six patients diagnosed with a liver tumor during a health check-up. 375 patients had incomplete blood tests or drank excessive alcohol in the following months. The remaining 545 patients were included and analyzed in this study. The inclusion criteria were patients who: (1) underwent abdominal sonography to evaluate NAFLD within 6 months prior to or after the cardiac MDCT evaluation; and (2) completed blood tests including liver enzymes (aspartate aminotransferase [AST] and alanine aminotransferase [ALT]), lipid profiles (total cholesterol, low-density lipoprotein cholesterol [LDL-C] and high-density lipoprotein cholesterol [HDL-C] and triglycerides), uric acid and glycated hemoglobin (HbA1c) after at least a 12-h fasting period before abdominal sonography. The exclusion criteria were patients who: (1) had evidence of viral hepatitis according to their medical records; (2) had autoimmune hepatitis; (3) were taking medicine which could have affected the diagnosis of steatohepatitis (such as corticosteroids, amiodarone, methotrexate and estrogen); (4) drank an excessive amount of alcohol, defined as alcohol consumption >20 g/day in males or >10 g/day in females; (5) had a past history of acute myocardial infarction, angina, transient ischemic attack or cerebrovascular disease; (6) had intrahepatic tumors or unconfirmed liver nodules; and (7) had incomplete blood laboratory test data for the recent 6 months.

### 2.2. Medical History Records and Laboratory Assessments

All basic data were acquired from the database of the health examination center of Asia University Hospital. A medical history of myocardial infarction, cerebrovascular disease, hypertension, diabetes mellitus, hyperlipidemia, viral hepatitis, and autoimmune hepatitis, and details of personal medicine prescriptions, smoking and alcohol consumption habits were collected by questionnaires before a general examination.

Body weight, height, and blood pressure were assessed on the cardiac evaluation day. Hypertension was defined as a self-reported history of hypertension, the use of antihypertensive medication, or a blood pressure of ≥140/90 mmHg. Diabetes was defined as a self-reported history of diabetes, the use of oral antidiabetic agents, or HbA1c > 6.5%. Hyperlipidemia was defined as a self-reported history of hyperlipidemia and the use of statins. Blood samples were collected after at least a 12-h fasting period, and laboratory data including AST, ALT, total cholesterol, HDL-C, LDL-C, triglycerides, uric acid and HbA1c were recorded.

### 2.3. Hepatic Ultrasonography

An abdominal ultrasound was used to assess the presence and severity of NAFLD using a high-resolution B-mode ultrasound machine (3.5 MHz, Voluson S6; GE Healthcare, Wauwatosa, WI, USA). The ultrasound was performed by experienced hepatobiliary gastroenterologists at Asia University Hospital. The diagnosis and severity of NAFLD followed the standardized criteria of ultrasonography classification [12,13] as follows:(1)Absent: normal echotexture of the liver was revealed without different echogenicities between liver parenchyma and renal cortex.(2)Mild: a slight increase in the echogenicity of the liver with normal visualization of intrahepatic vessels and diaphragm, and a mild difference in echogenicity between liver parenchyma and renal cortex.(3)Moderate: a moderate increase in the echogenicity of the liver with impaired appearance of intrahepatic vessels and diaphragm, and a difference in echogenicity between liver parenchyma and renal cortex.(4)Severe: a significant increase in the echogenicity of the liver with poor visualization of intrahepatic vessels and diaphragm combined with a significant difference in echogenicity between liver parenchyma and renal cortex.

### 2.4. Coronary Artery Calcium Score by Cardiac Multidetector Computed Tomography

The CAC score was measured using a 128-slice MDCT scanner (GE Healthcare Revolution EVO CT Scanner). The scanning parameters of the 128-slice MDCT were prospective EKG gating (120 kV tube voltage, 180–270 mA tube current and 2.5-mm slice thickness). Calcification in the coronary arteries was defined as a high-density lesion >130 HU according to the Agatston method. The parameters of all images were calculated and summed using a workstation (AW VOLUMESHARE 7, GE MEDICAL SYSTEMS SCS, Buc, Yvelines, France). Separate CAC scores were recorded on medical charts, and the total CAC score was categorized into four stages as described by Agatston et al. [14]. CAC stage-1 was defined as a total CAC score of 0, CAC stage-2 as a total CAC score between 1–99, CAC stage-3 as a total CAC score between 100–399, and CAC stage-4 as a total CAC score of ≥400.

### 2.5. Statistical Analysis

All of the patients’ basic characteristics, medical history and stage of coronary artery calcification were presented as percentages, and the blood sample biomarkers were presented as mean ± standard difference. According to the results of abdominal sonography, the severity of NAFLD was classified into four groups: a control group (non-NAFLD), mild NAFLD group, moderate NAFLD group, and severe NAFLD group. We used the chi-square test for categorical variables, such as the severity of NAFLD with sex, cardiovascular risk factors, CAC score > 0 and the presence of specific coronary calcification, and ANOVA for other continuous variables. Multivariate logistic regression analysis was used to assess associations between groups with CAC (CAC score > 0) and involved specific coronary calcification while controlling for potential confounding variables. Covariates in the multivariate model were chosen for clinical importance with previous statistical significance including age, sex, AST and ALT, triglycerides, HDL-C and uric acid. Significance was defined as *p* < 0.05. The correlation between CAC stage and severity of NAFLD was determined by Spearman correlation coefficient analysis. All data were analyzed using IBM SPSS version 25 (IBM Corp., Armonk, NY, USA).

## 3. Results

A total of 545 patients (mean age: 54 years) met the inclusion criteria and were enrolled in this study. Table 1 shows the characteristics of the study participants. Overall, 30.8% of the patients had a history of hypertension, 11.7% had diabetes, and 20.4% had hyperlipidemia. 

Of the 545 enrolled patients, 437 (80.2%) had ultrasonography-diagnosed NAFLD, and the prevalence was higher than in a previous Taiwanese study [15]. 188 patients were diagnosed as NAFLD with normal BMI value (BMI < 25 kg/m^2^), 191 patients were diagnosed as NAFLD with over-weight BMI (BMI between 25–29.9 kg/m^2^), and 58 patients were detected with NAFLD with obese BMI (BMI ≥ 30). 

Table 2 shows comparisons of the parameters between the different NAFLD groups. Male gender had a higher prevalence in moderate and severe NAFLD. Obesity also affected the presence of NAFLD, more severe NAFLD groups had a higher average body weight and body mass index. Not all of the traditional metabolic syndrome risk factors were associated with the severity of NAFLD. There were significant differences in systolic blood pressure, diastolic blood pressure, HbA1c, triglycerides, HDL-C and uric acid between the four NAFLD groups, however there were no significant differences in total cholesterol and LDL-C showed. High liver enzyme (AST and ALT) levels were associated with the severity of NAFLD, however the average value in each groups did not exceed the normal upper limit.

Of all patients, 242 (44%) had coronary artery calcification (CAC > 0). In addition, the proportion of patients with coronary artery calcification increased with the increased severity of NAFLD, and the chi-square test showed a significant difference between groups (*p* < 0.001). We also analyzed coronary artery calcification separately in the different severity NAFLD groups. The left anterior descending artery was the most commonly involved coronary artery in this study, and the highest proportion was observed in the severe NAFLD group. Moreover, the severe NAFLD group had a higher proportion of coronary artery calcification, especially in the left main artery, left anterior descending artery and right coronary artery.

In multivariate logistic regression analysis, a greater severity of NAFLD was associated with the presence of coronary artery calcification (odds ratio [OR] = 1.41, 95% CI = 1.13–1.77, *p* = 0.003) (Table 3). After adjusting for age, sex, AST and ALT, TG, HDL-C and uric acid, the risk of developing coronary artery calcification was 1.36-fold greater in the patients with different severity of NAFLD compared to those without NAFLD (OR = 1.36, 95% CI = 1.07–1.77, *p* = 0.016). Percentage of patients presenting of coronary calcification was higher in more severe NAFLD (Figure 1) and a positive correlation between CAC stage and severity of NAFLD was shown in Spearman correlation coefficient analysis (r_s_ = 0.204, *p* < 0.001). The highest OR for separate coronary artery calcification was 1.98 (OR = 1.98, 95% CI = 1.37–2.87, *p* < 0.001) in the left main artery, and the risk was still 1.71-fold greater after adjustments (OR = 1.71, 95% CI = 1.16–2.54, *p* = 0.007). In the adjusted module, the risk of left anterior descending artery calcification was 1.34 (OR = 1.34, 95% CI = 1.04–1.72, *p* = 0.025), the risk for right coronary artery calcification was 1.39 (OR = 1.39, 95% CI = 1.05–1.84, *p* = 0.021), and the risk for left circumflex artery calcification was 1.41 (OR = 1.41, 95%CI = 1.02–1.95, *p* = 0.039).

## 4. Discussion

Subclinical atherosclerosis is defined as an important arterial pathologic deformity which can further progress to major cardiovascular disease in asymptomatic patients. It can be detected by carotid intima-media thickness (CIMT), carotid-femoral pulse velocity and CAC score. The CAC score is a widely used and non-invasive technique with high sensitivity and specificity in predicting coronary artery disease and cardiovascular mortality compared with traditional metabolic scores [16]. According to the guidelines from the American College of Cardiology and American Heart Association (ACC/AHA), coronary artery calcium score (CAC score) was appropriate in patients with intermediate traditional CV risk [17]. The scores of 0, 1 to 100, 101 to 400, greater than 400 represented as stage normal, mild, moderate, and severe CAD risk. This re-classification could provide a reliable predictor for long-term all-cause mortality and cardiovascular mortality [18]. Even though the cost and radiation exposure of cardiac MDCT are concerns, CAC still has the ability to detect asymptomatic atherosclerosis at an early stage [19]. In an observational study, Budoff et al. reported that the risk of further coronary artery disease and all-cause mortality were significantly higher in patients with a CAC score > 0 than in the control group [20].

A retrospective study including 1173 asymptomatic population had identified that NAFLD is an independent risk factor for CAC progression, irrespective of the presence of metabolic syndrome [21]. Even though previous studies indicated that presence of NAFLD was an independent metabolic risk factor of coronary artery calcification, but the subgroup analysis demonstrated different results [22]. Kim et al.’s study suggested presence of NAFLD is more associated with occurrence of CAC in population without known cardiovascular risk factors, such as women, young age, normal-overweight, non-hypertensive, non-smoker, non-dyslipidemia, and non-diabetes [23]. Present study suggested the association between NAFLD and CAC is more significant in non-obese male participants [24]. According to our results, male participants have higher prevalence of moderate/severe NAFLD (42.4%, 10.2%) and higher staging of CAC (5.3%), and our results significantly identified NAFLD is an important metabolic manifestation responsible for the increased risk of CAC.

In our study, we demonstrated that the severity of NAFLD was associated with the presence of significant coronary artery calcification, especially in the left main coronary artery. Kim et al. reported that an increased CAC stage was associated with a higher prevalence of NAFLD (OR 1.84, 95% CI 1.61–2.10, *p* < 0.001) [23]. A recent meta-analysis which included 16 cross-section studies also showed that NAFLD was associated with CAC > 0 and CAC > 100, with pooled ORs of 1.41 (95% CI 1.26–1.57, *p* = 0.07, I^2^ = 66%) and 1.24 (95% CI 1.02–1.52, *p* = 0.1, I^2^ = 42%), respectively [25], which is consistent with our results. We also found an increasing prevalence of coronary artery calcification with the increased severity of NAFLD using sonographic image findings, which further confirms the relationship between NAFLD and CAC stage. The severity of NAFLD was weakly correlated with CAC stage in our results. In Kim et al.’s retrospective cohort study, they observed that a greater number of patients with NAFLD developed CAC in the following 4 years, and that NAFLD seemed to be a potential risk factor for the progression of coronary artery calcification [26]. Our study is valuable for keep following the individual cardiovascular outcome.

We then performed further analysis of the association between NAFLD and specific coronary calcification. In Koo et al.’s cohort study of 4186 patients, they found a positive association between NAFLD and calcification in multiple arteries. Furthermore, the patients with NAFLD had a higher risk of calcification in more than four arteries, and the results were not limited to coronary arteries, but also the carotid artery, thoracic aorta, celiac trunk, and superior mesenteric artery vascular beds [27]. Another study also reported a higher prevalence of coronary calcified plaques in patients with NAFLD, but no significant differences in high LDL, smoking status or metabolic syndrome in those with coronary calcified plaques [28]. Haddad et al. reported that patients with NAFLD had a higher risk of cardiovascular events in the further future, with a total relative risk (RR) of 1.77 (95% CI: 1.26–2.48, *p* < 0.001) compared with placebo, especially coronary artery disease (RR 2.26; 95% CI 1.04–4.92, *p* < 0.001) and ischemic stroke (RR 2.09; 95% CI 1.46–2.98, *p* < 0.001) [29]. According to the Agatston method, the weight of CAC in the left main coronary artery was equal to the weight of CAC in other coronary vessels; however, the presence of CAC in the left main coronary artery was strongly associated with adverse cardiovascular outcomes [30]. In our study, the severity of NAFLD was a good predictor for different coronary artery calcifications, especially in the left main coronary artery. Moreover, this result was still significant after adjusting for age, sex, TG, HDL-C and uric acid. Even though vessel-specific CAC scores are not yet used in current clinical guidelines, we still hypothesize that NAFLD is a risk factor for adverse cardiovascular events due to left main coronary artery calcification.

Even previous meta-analysis confirmed presenting NAFLD had a higher risk of cardiovascular events [29], recent study performed opposite conclusion. Liu et al. demonstrate severity of steatosis measured by controlled attenuation parameter (CAP) did not correlate with the incidence of cardiovascular events [31]. This result was different from our study outcome. According to the study design, Liu et al. followed up for a median of 26 interquartile range (IQR) [19,20,21,22,23,24,25,26,27,28,29,30,31] months and recorded 65 (1.6%) cardiovascular events. The incidence of CV events was lower than previous meta-analysis (Targheret al. concluded 34,043 participants and approximately 2600 CVD outcomes (>70% CVD deaths) over a median period of 6.9 years) [11]. We believed that observation of cardiovascular events needed longer following period to prove its relationship with NAFLD. Otherwise, the study population of Liu et al. had elder age and higher BMI than our participants. The author also mentioned about anti-platelet therapy and statin agents might reduce further cardiovascular events. These characteristics might cause different outcome between severity of NAFLD and cardiovascular event. Our cross-section study just proved the correlation between severity of NAFLD and coronary calcium score (as a prediction tool of further cardiovascular event), and we still need longer time for following the incidence of CV event in the further study.

According to review articles [2,32], the prevalence of NAFLD in Asia was 25%, but 8–19% of the population was diagnosed with lean-NAFLD (NAFLD in population with BMI < 25 kg/m^2^), the author believed that lean-NAFLD is an important issue for Asian people. According to our result, about 80% of participants (437/545 participants) had ultrasonography diagnosed NAFLD and 43% of these patients (188/437 participant with NAFLD) had lean NAFLD. It means that we detected more lean-NAFLD patients by sonographic criteria in non-obese population. But our study noticed the severity of NAFLD increased linearly with BMI and levels of triglycerides, but the result did not relate to low-density lipoprotein cholesterol and total cholesterol. Lean-NAFLD may be exist in Asian group, but we believed that obesity and hypertriglyceridemia were still the most important risk factors for NAFLD in Taiwan.

Metabolic syndrome is an important cause of progressive atherosclerosis, and it affects carotid artery intima thickness and is also related to the progression of coronary atherosclerosis [33,34]. Our results suggest that older age and the presence of diabetes were cofactors of NAFLD and metabolic syndrome, and that both factors affect the presence of coronary artery calcification in both males and females. With regards to the risk factor of metabolic syndrome, a cross-sectional study in a Chinese population reported that patients with NAFLD had a significant higher level of total cholesterol, very low density lipoprotein (VLDL) and TG [35]. However, we did not find an association between total cholesterol or LDL-C and the severity of NAFLD in the present study, although we did find that the levels of TG and HDL-C were associated with the presence of NAFLD. Some studies have investigated the association between dyslipidemia and the presence of NAFLD, however the conclusions have been inconsistent [36,37,38,39]. The reason could be due to racial differences, high rates of statin use and small study populations. The effect of 3-hydroxy-3-methyl-glutaryl-coenzyme A reductase (HMG-CoA) inhibitors, statins, on lipid metabolic production and anti-inflammation seem to be beneficial for NAFLD, and several studies have reported the benefits of statin treatment [40,41]. The results of these studies showed improvements in the progression of hepatosteatosis and decrease in serum liver biological markers (AST and ALT), however there was still no evidence of improvements in echoic NAFLD image findings [40,41]. The limitation of this study is the lack of data on hyperlipidemia treatment, which could have provided more practical evidence on NAFLD.

Sonography is an operator-dependent technique which is affected by physician’s experience and subjective impressions of hepatic echogenicity. Differentiation from steatosis to non-alcoholic steatohepatitis (NASH) is an important issue in the clinical practice because patients with “advanced NAFLD” or “NASH” would have higher risk of progressive liver fibrosis, all-cause mortality, CV mortality and other liver morbidity [11,42]. Therefore, recent study reported different types of parameters to increasing the accuracy and detect the severity of NAFLD. Ballestri et al. have proposed an ultrasonographic fatty liver indicator (US-FLI), an image score ranging from 2–8 by the intensity of hepato-renal contrast, which could rule out a diagnosis of severe NASH when US-FLI < 4 [43]. Hamaguchi et al. used other sonographic scoring system to define hepatic steatosis and its relationship with metabolic syndrome [44]. Even though sonography is not a gold standard for establishing advanced liver fibrosis, but it is still a mature and effective quantitative tool for hepatic steatosis by well experienced operator. Our study analyzed the retrospective ultrasonographic records describing by the original criteria which performed by Saverymuttu et al. [13]. The criteria were widely accessed by general physicians, and our ultrasonographic findings were still valuable in differentiating from steatosis to nonalcoholic steatohepatitis (NASH) [45,46,47]. Our study design was limited by the initial health care setting to confidently exclude advanced fibrosis, but we still focused on the relationship between ultrasonographic finding and specific coronary calcification.

Liver biopsy was a gold standard for NAFLD and NASH diagnosis, but it is unavailable for screening of population in our hospital. Magnetic resonance imaging proton-density-fat-fraction (MRI-PDFF) and Transient elastography with controlled attenuation parameter (CAP) are new non-invasive techniques in steatosis quantification. They provide more reliable information than conventional ultrasonographic criteria. Recent studies have shown PDFF was correlated to histology-determined steatosis with well sensitivity and accuracy [48,49]. CAP provide another quantification skill in detecting significant hepatic steatosis, and recent meta-analysis has suggested that it could be considered as a non-invasive method substitute of liver biopsy [50]. Neither PDFF nor CAP was available in our hospital setting, and that would be our limitation in study design

Liver fibrosis is an important prognostic factor in NAFLD. multiple non-invasive assessment for diagnosis of NAFLD/NASH generally performed instead of liver biopsy in clinical practice. Several serum biomarkers, such as fibrosis Score 4 (FIB-4 score), AST-Platelet Ratio Index (APRI) or NAFLD fibrosis score, are first-line screening tools commonly used for defining disease progression in primary health care centers. These evaluations focus on the progression of NAFLD, NASH, liver fibrosis or cirrhosis according to their accuracy and limitation [51,52]. FIB-4 score was widely available by general practitioners because of high negative predictive values (>90%) for severe liver fibrosis. Lee J, et al.’s research reveal NAFLD with higher liver fibrosis biomarkers (FIB-4 score) is associated with CAC score and CAC progression [53], and the result showed advanced NAFLD could be a predictor of progressive coronary artery disease. Ballestri et al. reported that four liver fibrosis biomarkers, such as FIB-4, Forns index, NFS and HFS, were positively correlated to three different kinds of cardiovascular risk scoring system [54]. These biomarkers may be another quantitative technique to identify the correlation between liver fibrosis and substantial cardiovascular risk. Vibration-controlled transient elastography(VCTE) and magnetic resonance elastography (MRE) were also provide accurate diagnosis and prognosis [55], but they are not available in our hospital setting. Even though our study design didn’t include these non-invasive assessment indicators, but we believe the further study can focus on the relationship between several liver fibrosis indicators and coronary calcium scoring system.

## 5. Conclusions

Our findings provide evidence that the severity of NAFLD is associated with CAC stage and separate coronary calcification involvement. Associations between NAFLD and traditional cardiovascular risk factors, including hyperlipidemia, higher blood pressure and increasing HbA1c were also confirmed. Future studies on the association between NAFLD and other atherosclerosis diseases can provide more information about the extrahepatic effect of NAFLD in Asian populations.

## Figures and Tables

**Figure 1 medicina-57-00807-f001:**
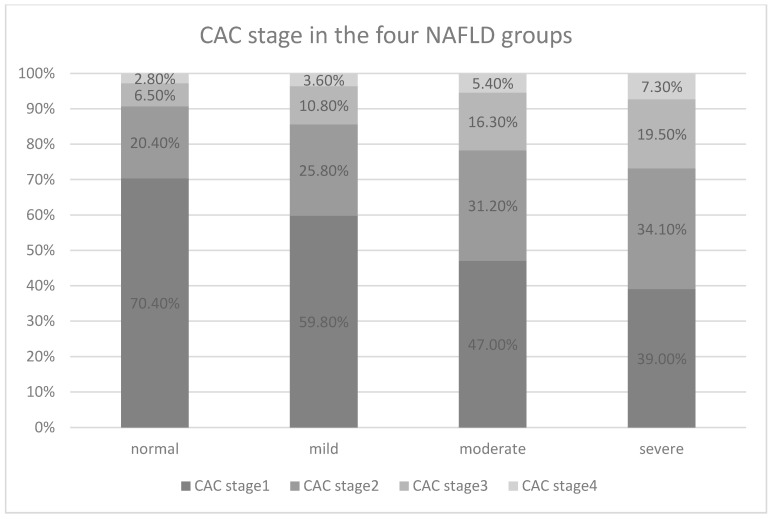
Percentage of CAC stage in different severity of NAFLD.

**Table 1 medicina-57-00807-t001:** Basic characteristic of the enrolled patients.

	Total	Males: 361 (66.2%)	Females: 184 (33.8%)
Age (years)	54.63 ± 10.60	53.69 ± 10.66	56.46 ± 10.26
Hypertension			
Yes	168 (30.8%)	127 (35.2%)	41 (22.3%)
No	377 (69.2%)	234 (64.8%)	143 (77.7%)
Diabetes			
Yes	64 (11.7%)	47 (13%)	17 (9.2%)
No	481 (88.3%)	314 (87%)	167 (90.8%)
Hyperlipidemia			
Yes	111 (20.4%)	79 (21.9%)	32 (17.4%)
No	434 (79.6%)	282 (78.1%)	152 (82.6%)
Height (cm)	164.90 ± 8.45	169.10 ± 6.33	156.68 ± 5.53
Weight (kg)	68.49 ± 13.32	73.60 ± 12.10	58.47 ± 9.30
BMI (kg/m^2^)	25.08 ± 3.73	25.72 ± 3.64	23.83 ± 3.60
SBP (mmHg)	128.15 ± 16.62	128.72 ± 3.64	127.03 ± 17.87
DBP (mmHg)	76.52 ± 12.06	79.92 ± 10.89	69.85 ± 11.47
AST (U/L)	23.26 ± 11.17	24.28 ± 11.25	21.25 ± 10.75
ALT (IU/L)	26.85 ± 19.07	30.80 ± 20.97	19.09 ± 11.19
HbA1c (%)	6.07 ± 1.06	6.16 ± 1.17	5.87 ± 0.79
Total cholesterol (mg/dL)	193.67 ± 40.28	192.2 ± 39.38	196.55 ± 41.96
HDL (mg/dL)	48.83 ± 25.48	45.54 ± 29.49	55.29 ± 12.50
LDL (mg/dL)	118.86 ± 35.37	119.43 ± 35.39	117.75 ± 35.42
TG (mg/dL)	152.80 ± 106.82	170.85 ± 120.57	117.40 ± 58.49
UA (mg/dL)	5.92 ± 1.47	6.46 ± 1.33	4.87 ± 1.14
NAFLD staging			
Normal	108 (19.8%)	54 (15%)	54 (29.3%)
Mild	194 (35.6%)	117 (32.4%)	77 (41.8%)
Moderate	202 (37.1%)	153 (42.4%)	49 (26.6%)
Severe	41 (7.5%)	37 (10.2%)	4 (2.2%)
CAC score			
1	303 (55.6%)	167 (46.3%)	136 (73.9%)
2	149 (27.3%)	116 (32.1%)	33 (17.9%)
3	69 (12.7%)	59 (16.3%)	10 (5.4%)
4	24 (4.4%)	19 (5.3%)	5 (2.7%)

Values are shown as mean ± standard (SD) deviation or number (%). BMI, Body Mass Index; SBP, Systolic blood pressure; DBP, diastolic blood pressure; AST, aspartate aminotransferase; ALT, alanine aminotransferase; HbA1c, glycated hemoglobin, HDL, High-density lipoprotein cholesterol; LDL, Low-density lipoprotein cholesterol; TG, triglyceride; UA, uric acid, NAFLD, non-alcoholic fatty liver disease; CAC, coronary artery calcification.

**Table 2 medicina-57-00807-t002:** Comparison of baseline characteristics between subjects in different NAFLD.

	Non-NAFLD (*n* = 108)	Mild NAFLD (*n* = 194)	Moderate NAFLD (*n* = 202)	Severe NAFLD (*n* = 41)	*p*-Value
Sex					<0.001
Male	54 (50%)	177 (60.3%)	153 (75.7%)	37 (90.2%)	
Female	54 (50%)	77 (39.7%)	49 (24.3%)	7 (9.8%)	
Age (years)	52.13 ± 11.39	55.64 ± 10.01	55.54 ± 10.57	51.88 ± 10.14	0.007
Weight (kg)	58.94 ± 9.43	65.62 ± 11.57	72.60 ± 10.37	87.05 ± 16.20	<0.001
Body mass index (kg/m^2^)	22.27 ± 2.87	24.29 ± 3.16	26.34 ± 3.03	30.04 ± 3.91	<0.001
Hypertension					0.003
Yes	24 (22.2%)	50 (25.8%)	76 (37.6%)	18 (43.9%)	
No	84 (77.8%)	144 (74.2%)	126 (62.4%)	23 (56.1%)	
Diabetes mellitus					0.001
Yes	7 (6.5%)	15 (7.7%)	32 (15.8%)	10 (24.4%)	
No	101 (93.5%)	179 (92.3%)	170 (84.2%)	31 (75.6%)	
Hyperlipidemia					<0.001
Yes	11 (10.2%)	34 (17.5%)	59 (29.2%)	7 (17.1%)	
No	97 (89.8%)	160 (82.5%)	143 (70.8%)	34 (82.9%)	
Systolic blood pressure (mmHg)	123.3 ± 18.44	127.92 ± 16.11	130.41 ± 15.74	130.93 ± 15.84	0.003
Diastolic blood pressure (mmHg)	72.21 ± 12.57	75.42 ± 11.68	78.66 ± 11.16	82.49 ± 12.45	<0.001
HbA1c (%)	5.78 ± 0.91	5.87 ± 0.80	6.25 ± 1.06	6.85 ± 1.06	<0.001
Total cholesterol (mg/dL)	191.35 ± 37.47	195.02 ± 41.59	195.10 ± 39.35	185.51 ± 45.62	0.691
Triglycerides (mg/dL)	101.47 ± 50.01	142.68 ± 88.64	175.61 ± 117.75	223.54 ± 159.81	<0.001
HDL-C (mg/dL)	54.80 ± 13.29	48.79 ± 12.03	47.26 ± 38.32	41.07 ± 10.47	0.014
LDL-C (mg/dL)	116.43 ± 32.63	118.76 ± 36.43	122.07 ± 34.95	109.98 ± 38.44	0.19
Uric acid (mg/dL)	5.32 ± 1.47	5.60 ± 1.44	6.42 ± 1.31	6.54 ± 1.40	<0.001
AST (U/L)	21.06 ± 8.04	21.23 ± 9.09	24.65 ± 12.29	31.73 ± 15.78	<0.001
ALT (IU/L)	19.34 ± 13.15	22.27 ± 17.20	30.91 ± 18.44	48.29 ± 23.30	<0.001
CAC score > 0					<0.001
Yes	32 (29.6%)	78 (40.2%)	107 (53%)	25 (61%)	
No	76 (70.4%)	116 (59.8%)	95 (47%)	16 (39%)	
LM involved					<0.001
Yes	4 (3.7%)	16 (8.2%)	30 (14.9%)	10 (24.4%)	
No	104 (96.3%)	178 (91.8%)	172 (85.1%)	31 (75.6%)	
LAD involved					0.001
Yes	26 (24.1%)	66 (34%)	90 (44.6%)	21 (51.2%)	
No	82 (75.9%)	128 (66%)	112 (55.4%)	20 (48.8%)	
LCX involved					0.01
Yes	10 (9.3%)	24 (12.4%)	42 (20.8%)	10 (24.4%)	
No	98 (90.7%)	170 (87.6%)	160 (79.2%)	31 (75.6%)	
RCA involved					0.001
Yes	19 (17.6%)	39 (20.1%)	55 (27.2%)	19 (46.3%)	
No	89 (82.4%)	155 (79.9%)	147 (72.8%)	22 (53.7%)	

Values are shown as mean ± standard (SD) deviation or number (%).BMI, Body Mass Index; SBP, Systolic blood pressure; DBP, diastolic blood pressure; AST, aspartate aminotransferase; ALT, alanine aminotransferase; HbA1c, glycated hemoglobin, HDL, High-density lipoprotein cholesterol; LDL, Low-density lipoprotein cholesterol; TG, triglyceride; UA, uric acid, NAFLD, non-alcoholic fatty liver disease; CAC, coronary artery calcification, LM, left main coronary artery; LAD, left anterior descending artery; LCX, left circumflex artery; RCA, right coronary artery.

**Table 3 medicina-57-00807-t003:** Multivariate analysis for different NAFLD with CAC > 0 and separated calcified coronary artery involvement.

	Model 1 Odds Ration (95% CI)	*p*-Value	Model 2 Odds Ration (95% CI)	*p*-Value	Model 3 Odds Ration (95% CI)	*p*-Value
CAC > 0	1.41 (1.13–1.77)	0.003	1.39 (1.09–1.77)	0.007	1.36 (1.07–1.77)	0.016
LM involved	1.98 (1.37–2.87)	<0.001	1.85 (1.26–2.7)	0.002	1.71 (1.16–2.54)	0.007
LAD involved	1.38 (1.10–1.73)	0.006	1.39 (1.09–1.77)	0.007	1.34 (1.04–1.72)	0.025
RCA involved	1.36 (1.06–1.74)	0.017	1.30 (1.00–1.69)	0.051	1.39 (1.05–1.84)	0.021
LCX involved	1.46 (1.09–1.97)	0.013	1.41 (1.03–1.93)	0.032	1.41 (1.02–1.95)	0.039

The severity of NAFLD was defined as a continuous variable. Model 1: adjusted for age and sex. Model 2: adjusted for age, sex, AST and ALT. Model 3: adjusted for age, sex, AST and ALT, TG, HDL-C and uric acid.

## Data Availability

Participants were enrolled from Asia university Hospital health examination center. The data presented in our study are available on request from the first and corresponding author. The data are not publicly available due to limitations of obtaining approval from the IRB for the disclosure of data. If anyone requires the data of this study, please to contact the corresponding author.

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
