# Peer review of "Association between the Severity of Nonalcoholic Fatty Liver Disease and the Risk of Coronary Artery Calcification"

_medicina, 2021, doi:10.3390/medicina57080807_

Round 1
Reviewer 1 Report
The manuscript has substantially improved.
Author Response
We thank reviewer for the comment.
Reviewer 2 Report
The number of patients with NAFLD is increasing worldwide and is an important public health concern.
Cardiovascular events are the most common cause of death in patients with NAFLD, and the relationship between NAFLD and coronary artery disease has received a great deal of attention.
In a cross-sectional study, the authors reported that NAFLD is associated with coronary artery calcification. This is a very interesting report. However, the following points are areas for improvement in this study.
First, in this study, the severity of NAFLD is classified by ultrasound findings. This seems to classify the severity of NAFLD according to the amount of fat in the liver, but it is subjective and may not be a widely used international classification method. Objective data is desirable. Liver biopsies would have been very useful data. Or, if CAP or PDFF had been recorded, the data would have been more reliable.
In addition, a large study of 5848 people reported that the amount of fat in the liver, as measured by CAP, did not correlate with the incidence of cardiovascular events (Am J Gastroenterol . 2017 Dec;112(12):1812-1823). This is inconsistent with this report and comments on this would be necessary.
Liver fibrosis is the most prognostic factor in NAFLD, and the incidence of cardiovascular events is also known to correlate well with liver fibrosis. Liver biopsy is still the most desirable, but the study would have been a more useful report if VCTE or MRE had been recorded.
Another small improvement would be to include an explanation or citation for the Agatston Method, as some readers may not be familiar with coronary artery disease.
Reviewer 3 Report
It is an interesting manuscript improving the knowledge related to NAFLD and other organ involvement. in the conclusion the authors mentioned that the data provide information for Asian population, so the authors should more clearly present data and/or specificities for Asian population. Do the authors have data regarding smoking? Waist circumference? It would be useful to have at least one noninvasive biomarker for NAFLD diagnosis.
Author Response
Review 3:
Q1: It is an interesting manuscript improving the knowledge related to NAFLD and other organ involvement. in the conclusion the authors mentioned that the data provide information for Asian population, so the authors should more clearly present data and/or specificities for Asian population. Do the authors have data regarding smoking? Waist circumference? It would be useful to have at least one noninvasive biomarker for NAFLD diagnosis.
Response 1:
We thank for reviewer's suggestion and add the following paragraph in discussion (page 9, 307-318) as following:
According to the review articles [1,2], the prevalence of NAFLD in Asia was 25%, but 8-19% of the population was diagnosed with lean-NAFLD (NAFLD in population with BMI< 25 kg/m2), the author believed that lean-NAFLD is an important issue for Asian people. According to our result, about 80% of participants (437/545 participants) had ultrasonography diagnosed NAFLD and 43% of these patients (188/437 participant with NAFLD) had lean NAFLD. It means that we detected more lean-NAFLD patients by sonographic criteria in non-obese population. But our study noticed the severity of NAFLD increased linearly with BMI and levels of triglycerides, but the result did not relate to low-density lipoprotein cholesterol and total cholesterol. Lean-NAFLD may be exist in Asian group, but we believed that obesity and hypertriglyceridemia were still the most important risk factors for NAFLD in Taiwan.
299 participants in our study were recorded with smoking history, the other 69 participants were with incomplete smoking records by retrospective data extraction. We can’t identify these participants were current smokers, non-smoker or ex-smokers. Most of our participants lost their waist circumference data in the medical records. We still enrolled all the patients because we believed these data was still valuable for analyzing the relationship between NAFLD and CAC. Incomplete data records were limitations of a retrospective cross-section study. Non-invasive biomarkers were useful quantitative techniques to identify the correlation between liver fibrosis and substantial cardiovascular risk. Even though our study design didn’t include these non-invasive assessment indicators, but we believe the further study can focus on the relationship between several liver fibrosis indicators in progressive NAFLD and coronary calcium scoring system.
- Younossi, Zobair M. “Non-alcoholic fatty liver disease - A global public health perspective.” Journal of hepatology vol. 70,3 (2019): 531-544. doi:10.1016/j.jhep.2018.10.033
- Li, J., et al., Prevalence, incidence, and outcome of non-alcoholic fatty liver disease in Asia, 1999-2019: a systematic review and meta-analysis. Lancet Gastroenterol Hepatol, 2019. 4(5): p. 389-398.
Round 2
Reviewer 2 Report
The content has been well revised. It is a scientifically important and well-written paper. I believe that this paper will bring great progress in the clinical practice of NAFLD.
This manuscript is a resubmission of an earlier submission. The following is a list of the peer review reports and author responses from that submission.
Round 1
Reviewer 1 Report
In the introduction, there is no hypothesis why NAFLD may be related to calcification of blood vessels?
a large sample tested is a great advantage of this project, but
the different number of patients in the materials and methods and the abstract is misleading, even if the description of exclusion from the study has been added. Add a sentence how many patients were included in the research right after "recruited 938 patients"
The topic has already been included in several scientific studies that were not included in the discussion
Association between noninvasive assessment of liver fibrosis and coronary artery calcification progression in patients with nonalcoholic fatty liver disease.
Lee J, Kim HS, Cho YK, Kim EH, Lee MJ, Bae IY, Jung CH, Park JY, Kim HK, Lee WJ. Sci Rep. 2020 Oct 27; 10 (1): 18323. doi: 10.1038 / s41598-020-75266-4.
PMID: 33110139
Non-alcoholic Fatty Liver Disease and Its Links with Inflammation and Atherosclerosis.
Abdallah LR, de Matos RC, E Souza YPDM, Vieira-Soares D, Muller-Machado G, Pollo-Flores P. Curr Atheroscler Rep. 2020 Feb 4; 22 (1): 7. doi: 10.1007 / s11883-020-0820-8. PMID: 32020371
Association between non-alcoholic fatty liver disease and coronary calcification depending on sex and obesity.
Kim SH, Park HY, Lee HS, Jung KS, Lee MH, Jhee JH, Kim TH, Lee JE, Kim HJ, Kim BS, Park HC, Lee BK, Choi HY. Sci Rep. 2020 Jan 23; 10 (1): 1025. doi: 10.1038 / s41598-020-57894-y. PMID: 31974458
What would the authors recommend regarding the results of their study?
best regards
Author Response
Q1: In the introduction, there is no hypothesis why NAFLD may be related to calcification of blood vessels?
Response 1:
We thank reviewer for the suggestion. We have added sentences in the introduction paragraphs on Page 2 and updated the references as following:
Studies have highlighted the association between NAFLD and coronary artery disease, and there were multiple underlying mechanisms overlapping between NAFLD and arterial calcification.[1, 2] [3] Stahl, E.P. et al.’s comprehensive review summarized the pathophysiological mechanisms of NAFLD increasing the risk of cardiovascular disease. It demonstrated six potential pathways, such as Systemic inflammation, endothelial dysfunction, hepatic insulin resistance, plaque formation, oxidative stress, and altered lipid metabolism, wound impact the risk of atherosclerosis, cardiovascular mortality in NAFLD population. [4]. Nevertheless, even meta-analysis revealed “significant NAFLD” with higher risk of incident of fatal and non-fatal CVD [5], but fewer study performed in the relationship between the severity of NAFLD and specific coronary artery calcification. Therefore, the aim of this study was to perform a cross-section analysis of the severity of NAFLD and its relationship with CAC score staging and separated calcified coronary artery involvement in an Asian population.”
Q2: Add a sentence how many patients were included in the research right after "recruited 938 patients"
Response 2:
We thank the pointing this out. We rewrite the section 2.1 study population in Material and Method on Page 2 as following:
There were 10 patients with positive finding for hepatitis B virus, 2 patients with positive finding for hepatitis C virus and 6 patients were diagnosed with a liver tumor during a health check-up. 375 patients were found with incomplete blood test or drank excessive alcohol in the following months. The remaining 545 patients were included and analyzed in this study.
Q3: The topic has already been included in several scientific studies that were not included in the discussion
Association between noninvasive assessment of liver fibrosis and coronary artery calcification progression in patients with nonalcoholic fatty liver disease.
Lee J, Kim HS, Cho YK, Kim EH, Lee MJ, Bae IY, Jung CH, Park JY, Kim HK, Lee WJ. Sci Rep. 2020 Oct 27; 10 (1): 18323. doi: 10.1038 / s41598-020-75266-4.
PMID: 33110139
Non-alcoholic Fatty Liver Disease and Its Links with Inflammation and Atherosclerosis.
Abdallah LR, de Matos RC, E Souza YPDM, Vieira-Soares D, Muller-Machado G, Pollo-Flores P. Curr Atheroscler Rep. 2020 Feb 4; 22 (1): 7. doi: 10.1007 / s11883-020-0820-8. PMID: 32020371
Association between non-alcoholic fatty liver disease and coronary calcification depending on sex and obesity.
Kim SH, Park HY, Lee HS, Jung KS, Lee MH, Jhee JH, Kim TH, Lee JE, Kim HJ, Kim BS, Park HC, Lee BK, Choi HY. Sci Rep. 2020 Jan 23; 10 (1): 1025. doi: 10.1038 / s41598-020-57894-y. PMID: 31974458
What would the authors recommend regarding the results of their study?
Response 3:
We thank for reviewer's suggestion and have added sentences in the discussion paragraphs on Page 9-10 as following:
In the recent study, non-invasive assessment for diagnosis of NAFLD/NASH performed instead of liver biopsy. Several biomarkers, such as fibrosis Score 4 (FIB-4 score), AST-Platelet Ratio Index (APRI) or NAFLD fibrosis score, are first-line screening tools commonly used for defining disease progression in primary health care centers. These evaluations focus on the progression of NAFLD, NASH, liver fibrosis or cirrhosis according to their accuracy and limitation. [6] [7] FIB-4 score was widely available by general practitioners because of high negative predictive values (>90%) for severe liver fibrosis. Lee J, et al.’s research reveal NAFLD with higher liver fibrosis biomarkers (FIB-4 score) is associated with CAC score and CAC progression[8], and the result showed advanced NAFLD could be a predictor of progressive coronary artery disease. Even though sonography is not a gold standard for establishing advanced liver fibrosis, but it is still a mature and effective quantitative tool for hepatic steatosis by well experienced operator.[9] Ultrasonographic findings were still valuable in differentiating from steatosis to nonalcoholic steatohepatitis (NASH) and finally fibrosis/cirrhosis [10, 11].Our study design was limited by the initial health care setting to confidently exclude advanced fibrosis, but we still focused on the relationship between ultrasonographic finding and specific coronary calcification.
A retrospective study, which included 1173 asymptomatic population, identified that NAFLD is an independent risk factor for CAC progression, irrespective of the presence of metabolic syndrome.[12] Even though previous studies indicated that presenting of NAFLD was an independent metabolic risk factor of coronary artery calcification, but the subgroup analysis demonstrates different results.[13] Kim et al.’s study suggested presenting of NAFLD is more associated with presence of CAC in population without known cardiovascular risk factors, such as women, young age, normal-overweight, non-hypertensive, non-smoker, non-dyslipidemia, and non-diabetes. [14]. Study of postmenopausal women demonstrates the relationship between NAFLD and CAC even after adjusting for well-established cardiovascular risk factors.[15]. Present study suggested the association between NAFLD and CAC is more significant in non-obese male participants [16]. According to our result, male participants have higher prevalence of moderate/severe NAFLD (42.4%, 10.2%) and higher staging of CAC (5.3%), and the result identified significant NAFLD in an important metabolic manifestation which increasing the risk of CAC and left main coronary artery calcification.
Reviewer 2 Report
The relationship of NAFLD and coronary calcification has been well studied. The paper does not provide additional knowledge to what we have known. Sonography is not good way to grade the severity of hepatic steatosis. it is affected by operators, and the quantification is very subjective. The methodology is fundamentally flawed.
Author Response
Reviewer 2
The relationship of NAFLD and coronary calcification has been well studied. The paper does not provide additional knowledge to what we have known. Sonography is not good way to grade the severity of hepatic steatosis. it is affected by operators, and the quantification is very subjective. The methodology is fundamentally flawed.
Response:
Thank for your professional suggestion and comment. Our study design was limited by the initial health care setting and might not able to confidently exclude advanced fibrosis. However, we have focused on the relationship between ultrasonographic finding and specific coronary calcification. Although ultrasonographic steatosis finding is not a gold standard for establishing advanced liver fibrosis, it is still a mature and effective quantitative tool for hepatic steatosis by well experienced operators.[9]. Ultrasonographic findings have been recognized as valuable reference in differentiating from steatosis to nonalcoholic steatohepatitis (NASH) and finally fibrosis/cirrhosis. [10, 11] Our purpose of this study was to identify the severity of NAFLD and its relationship with CAC score staging and separated calcified coronary artery involvement.
We believe that the presence of CAC in the left main coronary artery was strongly associated with adverse cardiovascular outcomes.[17] Further studies are in our list and will be done in the near further. More cautious designs regarding the methodology will be carefully addressed. Thanks for your advice.
References
- Gaggini, M., et al., Non-alcoholic fatty liver disease (NAFLD) and its connection with insulin resistance, dyslipidemia, atherosclerosis and coronary heart disease. Nutrients, 2013. 5(5): p. 1544-60.
- Lee, M.K., et al., Higher association of coronary artery calcification with non-alcoholic fatty liver disease than with abdominal obesity in middle-aged Korean men: the Kangbuk Samsung Health Study. Cardiovasc Diabetol, 2015. 14: p. 88.
- Juárez-Rojas, J.G., et al., Fatty liver increases the association of metabolic syndrome with diabetes and atherosclerosis. Diabetes Care, 2013. 36(6): p. 1726-8.
- Stahl, E.P., et al., Nonalcoholic Fatty Liver Disease and the Heart: JACC State-of-the-Art Review. J Am Coll Cardiol, 2019. 73(8): p. 948-963.
- Targher, G., et al., Non-alcoholic fatty liver disease and risk of incident cardiovascular disease: A meta-analysis. J Hepatol, 2016. 65(3): p. 589-600.
- Castera, L., M. Friedrich-Rust, and R. Loomba, Noninvasive Assessment of Liver Disease in Patients With Nonalcoholic Fatty Liver Disease. Gastroenterology, 2019. 156(5): p. 1264-1281.e4.
- Wong, V.W., et al., Noninvasive biomarkers in NAFLD and NASH - current progress and future promise. Nat Rev Gastroenterol Hepatol, 2018. 15(8): p. 461-478.
- Lee, J., et al., Association between noninvasive assessment of liver fibrosis and coronary artery calcification progression in patients with nonalcoholic fatty liver disease. Sci Rep, 2020. 10(1): p. 18323.
- Tanpowpong, N. and S. Panichyawat, Comparison of sonographic hepatorenal ratio and the degree of hepatic steatosis in magnetic resonance imaging-proton density fat fraction. J Ultrason, 2020. 20(82): p. e169-e175.
- Khodadoostan, M., et al., Comparison of liver enzymes level and sonographic findings value with liver biopsy findings in nonalcoholic fatty liver disease patients. Adv Biomed Res, 2016. 5: p. 40.
- Nelson, S.M., et al., Ultrasound Fatty Liver Indicator: A Simple Tool for Differentiating Steatosis From Nonalcoholic Steatohepatitis: Validity in the Average Obese Population. J Ultrasound Med, 2020. 39(4): p. 749-759.
- Cho, Y.K., et al., The impact of non-alcoholic fatty liver disease and metabolic syndrome on the progression of coronary artery calcification. Sci Rep, 2018. 8(1): p. 12004.
- Abdallah, L.R., et al., Non-alcoholic Fatty Liver Disease and Its Links with Inflammation and Atherosclerosis. Curr Atheroscler Rep, 2020. 22(1): p. 7.
- Kim, D., et al., Nonalcoholic fatty liver disease is associated with coronary artery calcification. Hepatology, 2012. 56(2): p. 605-13.
- Kim, M.K., et al., Association between nonalcoholic fatty liver disease and coronary artery calcification in postmenopausal women. Menopause, 2015. 22(12): p. 1323-7.
- Kim, S.H., et al., Association between non-alcoholic fatty liver disease and coronary calcification depending on sex and obesity. Sci Rep, 2020. 10(1): p. 1025.
- Lahti, S.J., et al., The association between left main coronary artery calcium and cardiovascular-specific and total mortality: The Coronary Artery Calcium Consortium. Atherosclerosis, 2019. 286: p. 172-178.
Reviewer 3 Report
GENERAL COMMENT
The Authors performed an interesting and well-written study showing that the ultrasonographic severity of NAFLD is associated to the risk of coronary artery calcification. Some comments may be raised at improving the manuscript.
SPECIFIC COMMENTS
- The Authors assessed the severity of NAFLD as grading the steatosis severity by ultrasonographic qualitative criteria (these reported in the review article by Ferraioli et. are generally those originally described by Saverymuttu et al BMJ 1986). We cannot know in the present study if patients with more severe NAFLD (ultrasonographic steatosis) had more NASH/fibrosis.
Current evidence suggest that liver fibrosis dictates the long-term course of NAFLD (Taylor et al, Gastroenterology 2020;158:1611–1625) and recent studies have shown that liver fibrosis biomarkers are associated with CAC and CVD risk in NAFLD patients (Lee J, Sci Rep. 2020 Oct 27;10(1):18323. doi: 10.1038/s41598-020-75266-4; Ballestri S, Diagnostics 2021 Jan 9;11(1):98. doi: 10.3390/diagnostics11010098.). Nevertheless, a recent metanalysis showed that subject with more 'severe' NAFLD (also assessed by US) had higher risk of incident of fatal and non-fatal CVD (Targher G, J Hepatol. 2016;65:589-600). Studies have found that ultrasonographic severity of NAFLD assessed semi-quantitatively reflects histological changes (Ballestri S, Liver Int. 2012 Sep;32(8):1242-52. doi: 10.1111/j.1478-3231.2012.02804.x.; Nelson SM, J Ultrasound Med. 2020 Apr;39(4):749-759. doi: 10.1002/jum.15154.)
Please comment and update literature.
If the Authors could assess non-invasively liver fibrosis this would be interesting.
Author Response
Reviewer 3
Q: The Authors assessed the severity of NAFLD as grading the steatosis severity by ultrasonographic qualitative criteria (these reported in the review article by Ferraioli et. are generally those originally described by Saverymuttu et al BMJ 1986). We cannot know in the present study if patients with more severe NAFLD (ultrasonographic steatosis) had more NASH/fibrosis.
Current evidence suggest that liver fibrosis dictates the long-term course of NAFLD (Taylor et al, Gastroenterology 2020;158:1611–1625) and recent studies have shown that liver fibrosis biomarkers are associated with CAC and CVD risk in NAFLD patients (Lee J, Sci Rep. 2020 Oct 27;10(1):18323. doi: 10.1038/s41598-020-75266-4; Ballestri S, Diagnostics 2021 Jan 9;11(1):98. doi: 10.3390/diagnostics11010098.). Nevertheless, a recent metanalysis showed that subject with more 'severe' NAFLD (also assessed by US) had higher risk of incident of fatal and non-fatal CVD (Targher G, J Hepatol. 2016;65:589-600). Studies have found that ultrasonographic severity of NAFLD assessed semi-quantitatively reflects histological changes (Ballestri S, Liver Int. 2012 Sep;32(8):1242-52. doi: 10.1111/j.1478-3231.2012.02804.x.; Nelson SM, J Ultrasound Med. 2020 Apr;39(4):749-759. doi: 10.1002/jum.15154.)
Response :
We thank for reviewer's suggestion and have added sentences in the discussion paragraphs on Page 9-10 as following:
In the recent study, non-invasive assessment for diagnosis of NAFLD/NASH performed instead of liver biopsy. Several biomarkers, such as fibrosis Score 4 (FIB-4 score), AST-Platelet Ratio Index (APRI) or NAFLD fibrosis score, are the first-line screening tools of disease progression in primary health care center. These evaluations are focused on the progression of NAFLD, NASH, liver fibrosis or cirrhosis according to their accuracy and limitation. [6] [7]
Our study design was limited by the initial health care setting, but we still focused on the relationship between ultrasonographic finding and specific coronary calcification. Recent studies focused on the relationship between advanced liver fibrosis and CAC progression, and the result showed advanced NAFLD could be a predictor of progressive coronary artery disease.[8]. Even the sonographic criteria is not a gold standard for establishing advanced liver fibrosis, but it is still a mature and effective quantitative tool for hepatic steatosis by well experienced operator.[9] Ultrasonographic findings were still valuable in differentiating from steatosis to nonalcoholic steatohepatitis (NASH) and finally fibrosis/cirrhosis [10, 11]. Further study which focused on liver fibrosis and coronary plaque will be interesting and valuable for CAD progression prediction. Thanks for your professional comments and constructive suggestions.
References
- Gaggini, M., et al., Non-alcoholic fatty liver disease (NAFLD) and its connection with insulin resistance, dyslipidemia, atherosclerosis and coronary heart disease. Nutrients, 2013. 5(5): p. 1544-60.
- Lee, M.K., et al., Higher association of coronary artery calcification with non-alcoholic fatty liver disease than with abdominal obesity in middle-aged Korean men: the Kangbuk Samsung Health Study. Cardiovasc Diabetol, 2015. 14: p. 88.
- Juárez-Rojas, J.G., et al., Fatty liver increases the association of metabolic syndrome with diabetes and atherosclerosis. Diabetes Care, 2013. 36(6): p. 1726-8.
- Stahl, E.P., et al., Nonalcoholic Fatty Liver Disease and the Heart: JACC State-of-the-Art Review. J Am Coll Cardiol, 2019. 73(8): p. 948-963.
- Targher, G., et al., Non-alcoholic fatty liver disease and risk of incident cardiovascular disease: A meta-analysis. J Hepatol, 2016. 65(3): p. 589-600.
- Castera, L., M. Friedrich-Rust, and R. Loomba, Noninvasive Assessment of Liver Disease in Patients With Nonalcoholic Fatty Liver Disease. Gastroenterology, 2019. 156(5): p. 1264-1281.e4.
- Wong, V.W., et al., Noninvasive biomarkers in NAFLD and NASH - current progress and future promise. Nat Rev Gastroenterol Hepatol, 2018. 15(8): p. 461-478.
- Lee, J., et al., Association between noninvasive assessment of liver fibrosis and coronary artery calcification progression in patients with nonalcoholic fatty liver disease. Sci Rep, 2020. 10(1): p. 18323.
- Tanpowpong, N. and S. Panichyawat, Comparison of sonographic hepatorenal ratio and the degree of hepatic steatosis in magnetic resonance imaging-proton density fat fraction. J Ultrason, 2020. 20(82): p. e169-e175.
- Khodadoostan, M., et al., Comparison of liver enzymes level and sonographic findings value with liver biopsy findings in nonalcoholic fatty liver disease patients. Adv Biomed Res, 2016. 5: p. 40.
- Nelson, S.M., et al., Ultrasound Fatty Liver Indicator: A Simple Tool for Differentiating Steatosis From Nonalcoholic Steatohepatitis: Validity in the Average Obese Population. J Ultrasound Med, 2020. 39(4): p. 749-759.
- Cho, Y.K., et al., The impact of non-alcoholic fatty liver disease and metabolic syndrome on the progression of coronary artery calcification. Sci Rep, 2018. 8(1): p. 12004.
- Abdallah, L.R., et al., Non-alcoholic Fatty Liver Disease and Its Links with Inflammation and Atherosclerosis. Curr Atheroscler Rep, 2020. 22(1): p. 7.
- Kim, D., et al., Nonalcoholic fatty liver disease is associated with coronary artery calcification. Hepatology, 2012. 56(2): p. 605-13.
- Kim, M.K., et al., Association between nonalcoholic fatty liver disease and coronary artery calcification in postmenopausal women. Menopause, 2015. 22(12): p. 1323-7.
- Kim, S.H., et al., Association between non-alcoholic fatty liver disease and coronary calcification depending on sex and obesity. Sci Rep, 2020. 10(1): p. 1025.
- Lahti, S.J., et al., The association between left main coronary artery calcium and cardiovascular-specific and total mortality: The Coronary Artery Calcium Consortium. Atherosclerosis, 2019. 286: p. 172-178.
Round 2
Reviewer 3 Report
The literature about serum non invasive biomarkers of fibrosis and ultrasound has not been fully updated according to the reviewer's suggestion.